# Acute Stress and Autoimmune Markers: Evaluating the Psychoneuroimmunology Axis in Firefighter Recruits

**DOI:** 10.3390/ijms26093945

**Published:** 2025-04-22

**Authors:** Andrea Schmitt, Nathan Andrews, Krista Yasuda, Mitchell Hodge, Rebecca Ryznar

**Affiliations:** 1College of Osteopathic Medicine, Rocky Vista University, Englewood, CO 80112, USA; andrea.schmitt@co.rvu.edu (A.S.); nathan.andrews@co.rvu.edu (N.A.); krista.yasuda@co.rvu.edu (K.Y.); mitchell.hodge@co.rvu.edu (M.H.); 2Department of Biomedical Sciences, College of Osteopathic Medicine, Rocky Vista University, Englewood, CO 80112, USA

**Keywords:** acute stress, first responders, autoimmunity, autoimmune mechanisms, molecular biomarkers, cortisol, C-reactive protein, complement C4, pigment epithelium derived factor, serum amyloid P

## Abstract

Chronic psychological stress is known to influence immune function and contribute to development of autoimmune disorders through dysregulated inflammatory responses. This study investigates relationships between acute stress, life trauma, and autoimmune salivary biomarkers in firefighter recruits during psychophysical stress training. Salivary samples were collected from firefighter recruits during two stress tests to evaluate responses to acute stress. Samples were obtained at three time points—pre-stress, post-stress, and recovery—across both tests. Cortisol was measured to characterize acute stress response (ASR) profiles, while immune function was assessed through the analyzing C-reactive Protein (CRP), Complement C4 (C4), Pigment Epithelium Derived Factor (PEDF), and Serum Amyloid P (SAP). Results showed significant changes in CRP, C4, and PEDF after stress inoculation. Higher previous life trauma was associated with lower baseline cortisol (r = −0.489) and delay in cortisol recovery (r = 0.514), suggesting a learned biological response, potentially protective against stress-induced dysregulation. Cluster analysis revealed four distinct cortisol ASR profiles which were found to have significantly different past life trauma (*p* = 0.031). These findings suggest that trauma history influences stress biomarker dynamics, potentially reflecting individualized adaptive or maladaptive responses. The insights gained may inform strategies to enhance stress resilience and mitigate autoimmune risk among high-stress populations.

## 1. Introduction

First responders are subjected to significant levels of acute and chronic stress, necessitating a comprehensive understanding of the factors contributing to stress for effective management strategies. The intensity, duration, and accumulation of stressors can vary significantly among individuals, highlighting the importance of personalized approaches to mitigate their impact over time. Chronic stress threatens and disrupts an individual’s state of homeostasis [1], tipping the scale to dysregulation and predisposing susceptible individuals to poor health outcomes and psychologic stress conditions such as post-traumatic stress disorder (PTSD) [2,3,4]. Factors such as prior life experience, previous traumas, specific environment, and genetics all affect how an individual responds to stress [5]. This individualized nature therefore warrants investigation into stress mechanisms and stress responses in first responders.

Psychological stress is a known, key modulator of immune function. Through the psychoneuroimmunology axis, acute and chronic stress activate hormonal and inflammatory pathways, linking psychological and physiological responses. In the presence of an acute stressor, the sympathetic nervous system (SNS) is activated to release catecholamines from the adrenal medulla [1]. With ongoing stress and SNS activation, the hypothalamus–pituitary–adrenal (HPA) axis is stimulated and leads to the release of cortisol from the adrenal cortex. In physiologic conditions, cortisol plays a role in the regulation of the immune system [5]. However, chronic and traumatic stress can lead to the dysregulation of the HPA axis through glucocorticoid resistance [6,7], and shift the immune response into a chronic state of inflammation [8,9,10]. HPA axis dysregulation and inflammatory states have been linked to the development and exacerbation of autoimmune disorders [11,12,13,14,15,16,17,18,19,20]. Furthermore, early life adversity can cause changes in target gene expression through alterations in epigenetic patterns [21,22], which can influence behavior, physiological outcome, and disease risk [21]. Studies have shown that individuals with PTSD have unique gene expression patterns involved in immune activation [23,24,25]. For example, experiencing trauma, and its related pathology, such as in the case of PTSD, leads to methylation at the klotho longevity locus [26] and the C reactive protein-associated AIM2 locus (Absent in Melanoma 2) [27,28]. Epigenetic changes at these loci promote cellular aging and inflammation [26,27,28] and are implicated in autoimmune conditions [29].

Regulators and indicators of the psychoneuroimmunology axis include a plethora of different biomolecular markers [30]. Those investigated in this study include C-reactive Protein (CRP), Complement C4 (C4), Pigment Epithelium Derived Factor (PEDF), and Serum Amyloid P (SAP). These four biomarkers have many shared roles, including a sensitivity to psychological stress [5,31,32,33,34,35,36,37,38,39], and are further implicated and involved in the regulation and modulation of inflammatory pathways [10,30,40,41,42]. Dysregulation of these biomarkers may predispose individuals to further pro-inflammatory cytokine release, tissue damage, and failure to clear apoptotic debris and pathogens [42,43,44]. This has been shown to lead to chronic lymphoid and myeloid cell activation, maturation of self-reactive B-cells, and spontaneous formation of germinal centers [44], predisposing individuals to developing self-antigens and autoimmune disorders [43]. Dysregulation of CRP, C4, PEDF, and SAP has been linked to several cases of autoimmune disease including rheumatoid arthritis (RA) [43,45,46], systemic lupus erythematosus (SLE) [43,47,48,49,50,51,52], systemic scleroderma [53,54], and type 1 diabetes mellitus [55,56,57,58], among others [43,59,60,61,62], emphasizing the interconnectedness of stress, immune regulation and modulation, and autoimmune disease pathogenesis.

First responders are often exposed to traumatic events, leading to high levels of acute and chronic stress, which may predispose them to immune dysregulation and autoimmune disorders. Investigating the relationship between acute stress responses, immune markers, and individual life trauma, is essential. Identifying potential adaptive and maladaptive patterns in this population may further lead to the development and implementation of therapeutic interventions. The aim of this study is to investigate the correlations between autoimmune biomarkers (C4, CRP, PEDF, and SAP) and acute stress responses, examine their associations with life trauma, and evaluate their roles in stress-adaptive immune mechanisms. Additional aims of this study include investigating cortisol fluctuations and the effects of prior life trauma on the ASR. This research has the potential to deepen our understanding of the biological mechanisms underlying stress and immune function, providing valuable insights for developing targeted interventions to enhance resilience and health outcomes in high-stress occupations, such as firefighting, while also informing broader strategies to support stress management and well-being in the general population.

## 2. Results

This study consisted of a cohort of 26 participants. Each participant completed a demographics survey to include questions regarding gender, age, race/ethnicity, prior diagnosis of chronic or autoimmune conditions and PTSD, active use of immunosuppressants and hypnotics, and prior experience as an emergency medical technician (EMT) or in the military. Demographics data are displayed in Appendix A. Twenty-five of our participants were biologically male and one was biologically female. Seventeen of our participants were ages 19–29, eight were ages 30–39, and one reported an age greater than 40. All our participants self-identified as white, with two reporting a Hispanic ethnicity. In total, 4 members had prior military experience and nearly half, 12, had prior EMT experience. Only one member had a diagnosis of PTSD, none had a prior diagnosis of chronic conditions or autoimmune disorders, and none were actively using immunosuppressants or hypnotics.

### 2.1. Salivary Cortisol Fluctuations Throughout Acute Stress and Recovery

Each of our participants underwent a simulation, aimed to stimulate an acute stress response, before and after their 16 weeks of training. Mean salivary cortisol aggregates for the three timepoints (T1, T2, T3) across both simulations, labeled A and B, respectively, are displayed in Figure 1. Prior to the 16-week training protocol, stress test A salivary cortisol levels displayed a statistically significant 7% increase between time points T1 and T2 (*p* = 0.004) and a significant 7% decrease between time points T2 and T3 (*p* = 0.026). After 16 weeks of training, participants displayed a statistically significant decrease in cortisol levels at T1 (*p* = 0.033). This is followed by a statistically significant decrease in cortisol levels at T2 (*p* = 0.005).

### 2.2. The Effects of Prior Life Trauma on the Acute Stress Response

Prior life trauma was correlated to a participants cortisol response curve, composed of cortisol levels at T1, during acute stress as calculated by the relative percentage change between T1 and T2, and recovery as calculated by the relative percentage change between T2 and T3 for both stress test A and stress test B, as displayed in Figure 2. At baseline (T1), participants displayed a negative correlation, r = −0.489, between prior life trauma and cortisol levels during stress test A, prior to 16-week fire academy training and a near-zero correlation, r = −0.096, and during stress test B, after 16-week fire academy training. During the acute stress time period, correlations of r = 0.043 and r = −0.197 were found to be the same during stress test A and B, respectively. Recovery time periods showed positive correlations between prior life trauma and cortisol response of r = 0.514 and r = 0.384 during stress test A and B, respectively.

### 2.3. Autoimmune Biomarkers in the Acute Stress Response

Four autoimmune biomarkers were collected from participants at the three timepoints across the two stress tests, A and B, as shown in Figure 3. Biomarkers C4, PED-F, and CRP displayed statistically significant changes between stress test A and B across various time points. C4 and PEDF displayed a significant difference between the two stress tests at time point T2, immediately after stress induction. CRP displayed significant differences at all three time points. SAP did not show significant differences at any time point.

Analysis was then conducted to compare a participant’s salivary cortisol level to their autoimmune biomarker level across both stress tests at three time points. Using Pearson correlation coefficients, these Life Events Checklist (LEC-5) scores were correlated with levels of CRP, C4, PEDF, SAP, and cortisol at each time point as well as the relative percentage change in biomarkers during acute stress and recovery time periods displayed in Figure 4.

### 2.4. Cluster Analysis as a Means to Group Participants’ Cortisol Response

As a proof-of-concept, cortisol levels from stress test A were used in a K-means cluster analysis to separate participants based on their response to acute stress without training. Four clusters were obtained, displayed in Figure 5, from analysis with the populations listed: Cluster 1: 7, Cluster 2: 4, Cluster 3: 4, Cluster 4: 11. Average LEC-5 scores for the four clusters are listed: 47, 19, 66, and 36, respectively. Further analysis of variances (ANOVA) between clusters revealed significant differences in LEC-5 scores, secondary to cortisol response curves of stress test A (*p* = 0.031). Comparing autoimmune data both inter- and intra-cluster did not reveal results of significance.

## 3. Discussion

First responders are often exposed to traumatic events, leading to high levels of acute and chronic stress, which predispose them to immune dysregulation and autoimmune disorders [8,9,11,12,13,14,15,16,17,18,19,20]. Investigating the relationship between acute stress responses, immune markers, and individual life trauma, is essential for identifying potential adaptive and maladaptive patterns in this population. This pilot study explored the interplay between acute stress responses, autoimmune salivary biomarkers, and psychosocial factors in firefighter recruits. This study seeks to advance our understanding of stress-induced immune variability in high-stress populations.

### 3.1. Salivary Cortisol Fluctuations Throughout Acute Stress and Recovery

An acute stress response was first established for stress test A and B using a known salivary stress marker, cortisol [39,63,64]. Our results showed that the acute stress response was present for both stress tests A and B. Additionally, there was a significant decrease during the prestress and stress phases of stress test B when compared to stress test A. This attenuated stress response may be a consequence of stress habituation. Habituation, defined as any decrease in responsiveness to a repeated stimulus, a form of non-associative learning [65], has been applied to acute stress and the HPA axis. Several animal and human studies have shown a decrease in HPA physiologic response following repeated acute stress exposures [65,66,67,68]. In addition to experiencing a degree of stress habituation, the recruits may have been more prepared, physically and psychologically, during the second stress test.

### 3.2. The Effects of Prior Life Trauma on the Acute Stress Response

Understanding stress reactions in relation to an individual’s prior life trauma may be important for understanding the development of certain stress-related disorders such as PTSD, depression, and anxiety. Thus, after establishing the acute stress response, we analyzed cortisol in relation to LEC-5 scores. Our results showed that recruits with high LEC-5 scores had a moderately lower stress level during prestress compared to those with low life trauma. The current literature supports the finding that individuals with greater lifetime stressor exposure, when faced with acute stress, are known to have blunted acute stress reactions [69,70,71,72].

Reactions to repeated stress exposures are individualistic and influenced by personal and environmental factors [73]. Regarding the HPA axis and cortisol reactivity to stress, individuals may respond via habituation or sensitization. In habituation, the cortisol stress reaction is decreased at the second acute stress exposure, whereas in sensitization, the cortisol stress reaction is increased [73]. Most healthy individuals are known to experience habituation, whereas 30–40% of individuals are non-habituators or experience sensitization [73]. Interestingly, when comparing stress test A to B, the moderate relationship between LEC-5 scores and cortisol, discussed above, was not seen again in stress test B. Specifically, individuals with low life trauma had a decrease in prestress cortisol levels from stress test A to B, whereas individuals with high life trauma showed similar prestress cortisol levels. These results suggest that individuals with lower life trauma experienced prestress habituation from test A to B. Hartwig et al., using the free energy principle, suggests that under physiologic conditions, stress habituation makes stress more tolerable and protects against the deleterious effects of toxic stress [74]. Furthermore, our results suggest that individuals with higher life trauma did not experience habituation to the second stress test. This result is supported by the current literature that suggests individuals with higher life stress experience poor habituation in response to stress [75,76].

During the recovery phase, individuals with higher life trauma had moderately increased cortisol levels after stress compared to those with lower life trauma. Supporting our result, one study showed that individuals who had higher cortisol levels following acute stress were associated with an increase in recent minor life stress [77]. Similarly, Elzinga et al. found that individuals with PTSD had elevated cortisol in stress recovery compared to controls following trauma exposure [78]. This cortisol pattern may be reflective of resilience patterns in individuals with numerous traumatic life experiences, as these events may be associated with diminished stress reactivity. A 2023 review of the acute stress response and resilience demonstrated a significant relationship between the acute cortisol response and resilience [79]. The majority of studies found a negative correlation [79]; however, one study supports an elevated cortisol level reflecting increased resiliency [80]. Furthermore, post-stress rumination is known to be associated with increased cortisol reactivity following acute stress [81], suggesting one psychological mechanism leading to increased cortisol levels in this population.

On the other hand, changes in cortisol stress reaction patterns may be associated with a maladaptive response to stress. Low basal cortisol because of HPA axis dysregulation may predispose individuals to increased stress vulnerability and predisposition to developing psychiatric disorders [82]. Taken all together, many variables including genetic vulnerability, previous stress experience, coping and personality styles [82], resiliency [79], and post-stress rumination [81] affect individual post-stress recovery patterns and specific mechanisms and consequences require further investigation.

Cluster analysis was used in this study for the purpose of exploring trends in individual immune profiles and as a proof-of-concept-technique. As such, cluster analysis is not statistically significant and limits causal inferences. Our results are interpreted with this in mind. Cluster analysis revealed four distinct cortisol ASR profiles secondary to cortisol response curves of stress test A. Further ANOVA between clusters revealed differences in LEC-5 scores (*p* = 0.031). These findings suggest that trauma history may be associated with stress, potentially reflecting individualized responses. There is substantial literature linking trauma history to acute stress responses. Some literature describes heightened stress responses [78] while others describe a blunted or diminished acute stress response [69,70,71,72,83,84] in relation to trauma exposure. Reactions to stress are individualized and highlight the importance of personalized approaches in understanding stress resilience. The future identification of biomarker patterns through statistically significant analysis may inform targeted interventions aimed at improving health outcomes in high-stress populations.

### 3.3. Autoimmune Biomarkers in the Acute Stress Response

Another main objective of this study was to evaluate changes in salivary autoimmune biomarkers (C4, PEDF, CRP, and SAP) during acute stress. Psychological stress is a known key modulator of immune function through the psychoneuroimmunology axis, and dysregulation has been linked to the development and exacerbation of autoimmune disorders. Alterations in C4, PEDF, CRP, and SAP have all been linked to stress states [5,10,31,32,33,34,35,36,37,38,85] and are further implicated and involved in the regulation and modulation of inflammatory pathways [10,30,40,41,42,86]. Specifically, C4, PEDF, CRP, and SAP participate in clearing apoptotic debris, leukocyte chemotaxis, macrophage activation, and cytokine secretion [42,43]. Chronic alterations or dysregulation in C4, PEDF, CRP, and SAP may predispose individuals to inflammation and tissue damage, leading to the development of self-antigens and autoimmunity [43]. Dysregulation of these biomarkers has been linked to cases of RA and SLE, among others [43,45,46,47,48,49,50,51,52,53,54,55,56,57,58,59,60,61,62,87]. Research regarding fluctuation in these biomarkers before, during, and after an acute physical and psychological stress, however, is limited and warrants investigation.

Our analysis of CRP, C4, PEDF, and SAP in association with Stress Tests A and B showed a general trend of the biomarkers. Biomarkers all increased during the stressful event and were followed by a decrease during recovery (Figure 3). These results suggest that CRP, C4, PEDF, and SAP levels may be influenced by acute physical and psychological stress. These results are supported by current literature, as inflammatory cytokines are known to fluctuate in response to psychological trauma [88,89].

When stress test B was compared to stress test A, CRP showed a significant decrease across prestress, stress, and recovery; C4 had significant decrease in stress and recovery and PEDF had a significant decrease in stress. These results suggest there was a decrease in inflammatory markers and pathways with habituation to stress. A recent systematic review [73] found that repeated stress exposure was associated with habituation of inflammatory markers in 22% of studies. Specifically, IL-6 was the most extensively studied cytokine [73]. Additional studies reported a lack of habituation or sensitization of immune markers in response to stress, highlighting the high degree of variability of these markers and lack of a definitive conclusion regarding the relationship of immune markers and habituation [73]. Furthermore, the inability to habituate to repeated stress has been linked to many health problems, including systemic inflammation [73]. In summary, stress habituation may be a potential protective factor against stress-induced dysregulation and thus the development of autoimmune disease.

Pearson correlation coefficients showed a significant or moderate relationship between CRP and SAP at all time points during both stress tests, likely due to their interrelated nature. CRP and SAP are opsonins belonging to the pentraxin family. CRP is also a major acute phase reactant in humans, used clinically as a marker of inflammation, and participates in the complement pathway, apoptosis, phagocytosis, cytokine production (IL-6 and TNF-α), and leukocyte chemotaxis [42]. CRP and SAP expression are primarily induced by pro-inflammatory cytokines IL-6 and IL-1 [86], which are induced by acute stress [90]. Further results showed a significant or moderate correlation between PEDF and CRP, and PEDF and C4 at several time points. Pigment epithelium-derived factor is a protease inhibitor belonging to the serpin family and has anti-angiogenic, anti-inflammatory, and antioxidant properties [41]. Our results are supported by the current literature as PEDF is known to be associated with inflammatory markers, including acute phase reactant CRP [91,92,93] and complement factor C4 [45]. PEDF, CRP, and C4 are known to be altered in situations involving acute stress. As a consequence of the relationships observed between CRP, SAP, PEDF, and C4, and their known associations with stress [10,15,32,33,34,35,36,37,38,85] and autoimmune conditions [43,45,46,47,48,49,50,51,52,53,54,55,56,57,58,59,60,61,62,87], these results may help define patterns of inflammation involved in the stress-induced autoimmune process.

Our Pearson correlation coefficient results additionally showed an increase in biomarker relationships in recovery when compared to stress in both stress tests. These results suggest that adaptive responses may emerge during recovery rather than during the stress event itself. Current literature shows that improved stress responses further improve the regulation of inflammatory pathways [94].

Although we link biomarkers to autoimmune risk, their significance remains unclear. The trends explored here should be interpreted with caution, as they may not reflect a direct or meaningful connection to autoimmune risk. However, the biomarker’s previous characterization as playing a role in autoimmune disease leads us to the tentative possibility that autoimmune risk could vary among individuals in high stress careers, though this is not a definitive conclusion. This area requires further investigation, as alternative explanations and interpretations of these trends cannot be ruled out. Specifically, alternative explanations could include a relationship better explained by a physiologic response to chronic stress; confounding third variables such as differences in sleep, nutrition, or underlying medical conditions; biologic and social factors such as gender, age, race or ethnicity; or occupational sample bias.

### 3.4. Limitations

This study has limitations that should be taken into consideration. The sample size in this pilot study is relatively small, at 26 participants, which limits statistical power and generalizability. The sample population was also considerably homogenous, with all participants being in the same career field and the majority of participants being under 40 years old (96.2%), Caucasian (100%), and with previous experience as first responders (61.5%). This limits the generalizability of the study, strength of the statistical analysis, and potential reproducibility of our results. Furthermore, 96.2% of the participants were male and only one study participant was female. This discrepancy is partly due to the nature of the firefighting occupation having unequal representation of males versus females. We acknowledge this as a limitation of our study as there are potential differences in stress responses between males and females that we were unable to assess for. Similarly, there are differences in stress responses between different races. These were variables that we were unable to assess given this study’s specific participant demographics. This study also used participants from a single fire academy. While this allowed us to perform controlled and standardized stress tests at the facility, it also limits external validity to fire departments with different training resources. Taken altogether, this study is limited by sample bias due to a small sample size and a homogenous population.

While our study screened for and excluded participants using immunosuppressants or hypnotics, we did not screen for any other medication use. We acknowledge that medications such as antidepressants, anxiolytics, and anti-inflammatories may interfere with our results. Future studies should consider the impact of these and other medications.

We were not able to control for the effects of circadian rhythm on cortisol and the ASR. However, our three-sample sequence across two stress tests was designed to assess each participant’s ASR at baseline, with the initial sample serving as the negative control. This study would benefit from the addition of a non-firefighter control group or the addition of different stress tests to better distinguish stress related specifically to firefighters. The addition of these control groups will be integrated into our team’s future directions.

Further factors such as hydration and nutrition, beyond fasting for 30 min prior to the stress test, may have varied between participants. Similarly, sleep patterns and chronic stress may vary. These factors were not controlled for and may have affected salivary samples or the ASR.

Additionally, we are aware of the potential for confounding variables (such as significant life events or increased familiarity in navigating dangerous fire training scenarios) that may have occurred in the recruits’ personal and professional lives during the time between stress test A and B as we did not readminister the LEC-5 survey before stress test B. Any major life events experienced during this time could impact the stress response of recruits. As such, our study is limited in longitudinal tracking of psychological state changes which should be addressed in future studies. Furthermore, we did not measure individual perceived levels of stress; however, we have determined that this training exercise does induce an acute stress response in this and previous studies [88,95]. Additionally, biomarkers were selected based on an available, representative panel of immune markers, and not all biomarkers involved in autoimmunity and/or the acute stress response were investigated. Although salivary sampling is minimally invasive, some studies have shown that measurement of salivary biomarkers may not reflect blood values or systemic inflammation [96,97]. Furthermore, our saliva collection method could introduce changes in pH which could result in potential errors or skewing of our results.

Finally, we recognize this study as a pilot study which solely seeks to identify relationships between the explored biomarkers. As such, some mild correlations are expected and are unable to be used to identify causal relationships. Further studies are needed to isolate and control for specific variables to attempt to identify causal relationships. Furthermore, cluster analysis provided utility for proof of concepts and exploring trends in biomarkers across individual profiles; however, this portion of the study’s reliance on cluster analysis also limits causal inferences. Considering the above limitations, this study may be limited in reproducibility and should be repeated considering these factors.

### 3.5. Future Directions

Several future directions for this study are considered. Addressing the major limitations of this study, further studies should include larger and more diverse populations of participants. The findings may have broader implications for stress-related health interventions in similarly vulnerable populations and should be further explored. Additionally, future directions will include the incorporation of non-firefighter control groups assessed utilizing the same stress test and procedure. Further studies should also incorporate different types of stress tests. Incorporating these controls will help researchers better investigate stress specific to firefighters. To improve the transparency of our study, future directions include improved longitudinal tracking of stress states and major life changes between stress tests. Furthermore, future studies could incorporate additional markers of autoimmune disease and explore previous diagnosis of autoimmune disorders in first responders undergoing stress testing. Trends identified through cluster analysis should be further expanded and researched in larger, longitudinal cohorts to validate our findings and to better understand acute stress, autoimmune biomarkers, and individual profiles.

## 4. Materials and Methods

### 4.1. Study Design

This is a prospective cohort study investigating levels of salivary immune biomarkers of participants at baseline, during, and after a psychophysical challenge with both emotionally and physically stressful components. Rocky Vista University College of Osteopathic Medicine approved the study through its IRB committee, which included extension and modification from a previous study (IRB# 2019-0092) [95]. The study was conducted in accordance with the Declaration of Helsinki. Sampling was conducted with South Metro Fire at the Metro Fire Training Center in Littleton, CO. All participants consented. Participants completed a demographics survey (Appendix A), Life Events Checklist (LEC-5) questionnaire [98,99] to assess life trauma, and provided salivary samples (described below).

### 4.2. Participants

Participants were selected based on their enrollment in the fire academy. The final sample size consisted of 26 fire recruits currently enrolled in the fire academy with South Metro Fire Department. Exclusion criteria included members who have already completed a fire academy and members not actively involved with the current South Metro Fire Academy. Consenting measures for participants were both written and verbal regarding measures of the study, and each written consent form was retained before participation in the study. Stress test 1 was performed in June 2022 and stress test 2 was performed in July 2022.

### 4.3. Demographic Data Collection

Demographic information was collected from each participant prior to the stress test including gender, age, race, and ethnicity (Appendix A). Additional information collected consisted of voluntary reporting of previous diagnosis of PTSD, previous diagnosis of chronic conditions or autoimmune disorders, active use of immunosuppressants and hypnotics, prior experience as an EMT or paramedic, and prior military experience. Exclusion criteria included any participants with chronic or autoimmune disorders and use of immunosuppressants or hypnotics. Use of antidepressants, anxiolytics, anti-inflammatory agents, and other medications were not assessed and participants using these medications were not excluded.

### 4.4. Life Events Checklist

Participants completed a life events checklist (LEC-5) survey (Appendix A), a life events checklist examining prior psychological traumas participants may have experienced [98,99]. After completing the LEC-5 questionnaire, consisting of 17 questions, a severity score of 1–3 was assigned, as described in Speakman et al. [88]. An event that happened to the participant was scored a 3, an event that was witnessed by the participant was scored a 2, an event that the participant learned about was scored a 1, and a traumatic event that does not apply to the participant was scored a 0.

### 4.5. Stress Test

All fire recruits participating in fire academy training must complete and pass a physical and psychological stress test, as previously described in Ryznar et al. [95]. Recruits are required to pass this test twice, at both the beginning and at the end of their fire academy to stay, in compliance with their respective fire departments. Ryznar et al. demonstrated that this standardized test does elicit an acute stress response [95]. Recruits were fully outfitted, blindfolded, and actively distracted while they navigated a simulated collapsing, overheating, multi-story building. Recruits were challenged to follow a fire hose to navigate the building. Their masks keep them in a low oxygen environment for the duration of the event, and the recruits were challenged to keep their masks on for the duration of the activity to pass the test. The event takes place for 10–15 min. This test was specifically designed to simulate real-life scenarios faced by firefighters and push the recruits to their stress limits to prepare them for real firefighting scenarios where factors such as limited visibility, limited oxygen, and distractions may be present. Furthermore, this training helps prepare recruits for the rigors of their highly demanding and stressful careers in firefighting.

### 4.6. Salivary Sample Collection

Three sets of saliva samples were collected from each participant during stress test A. Three additional sets of saliva samples were collected from each participant during stress test B. Sampling took place shortly before the stress test (“pre-stress”), immediately following the stress test (“post-stress”), and post-stress test (“recovery”). Participants were asked not to eat or drink for at least 30 min prior to the stress test. Saliva collection started at 8:00 am with the pre-stress samples and the stress test followed within two hours of the initial collection. Post-stress samples were collected immediately following completion of the event. Recovery samples were collected one hour after the event. Saliva collection utilized the whole stimulated saliva method, as outlined by Ryznar et al. [95]. Participants chewed sugar-free gum for five minutes, followed by collection of 1 mL of saliva in 1.5 mL Eppendorf tubes. Following collection, a protease inhibitor was added to each sample at a concentration of 1 μg/mL. All samples were immediately placed on ice to avoid protein degradation, then stored frozen and transported on dry ice to Eve Technologies (Calgary, AB, Canada). Concentrations of cortisol and autoimmune predisposition markers in each sample were measured by Eve Technologies using Millipore assays. Specific sample assays include the MILLIPLEX^®^ Human Circadian/Stress Magnetic Bead Panel (HNCSMAG-35K) and MILLIPLEX^®^ Human Neurodegenerative Disease Magnetic Bead Panel 2 (HNDG2MAG-36K), respectively. The assays use Luminex xMAP technology for multiplexed quantification of cortisol and biomarkers including CRP, C4, PEDF, and SAP. Eve technologies determined the concentration of each biomarker based on an established standard curve, along with providing known sensitivities and specificities for each marker (Appendix A).

### 4.7. Statistical Analysis

Sample collection resulted in salivary samples from 26 fire academy recruits measuring cortisol, C4, CRP, PEDF, and SAP levels for three intervals at two separate stress tests. Analysis was conducted in RStudio version 2022.12.0+353. All reported concentrations with values outside the standard curve were scaled to the closest observed concentration within the standard curve. Due to the variability of our samples, each raw value was scaled using the common logarithm and a relative percentage change was calculated and utilized to represent biomarker fluctuation throughout the acute stress event. Descriptive statistics tables for each analyte at each time point are shown in Appendix A. Individual cortisol trends were used to quantify and assess the ability of the simulation to elicit a measurable acute stress response. Two-sided paired *t*-tests were used to evaluate significant cortisol differences between the two stress tests.

To assess the effects of prior life trauma on the individual acute stress response, a linear relationship was calculated. Prior life trauma, as quantified through the LEC-5 assessment, and cortisol levels at various time points, including baseline, during stress, and during recovery time points, were evaluated using Pearson correlation coefficients. After a quantifiable stress response had been reported, autoimmune biomarker trends were evaluated in a manner similar to cortisol, using two-sided paired *t*-tests to evaluate for differences between the two stress tests. A Pearson correlation coefficient matrix was then generated for both stress tests to evaluate relationships between all biomarkers.

As a proof-of-concept to evaluate for an autoimmune biomarker predictive value shaped by the acute stress response, a k-means cluster analysis was performed based on participants’ individual cortisol response curves during stress test A. The optimal number of clusters was calculated using a WSS (Within-Cluster-Sum of Squared Errors) score and using the silhouette method [100]. An ANOVA was completed on LEC-5 scores separated by cluster. These individual clusters were then re-evaluated for habituation after training using two-sided paired *t*-tests and Pearson correlation coefficients to evaluate relationships with autoimmune biomarkers.

## 5. Conclusions

This work provides novel insights into the psychoneuroimmunology of acute stress, emphasizing the dynamic interplay between immune and psychosocial factors in first responders. Our findings demonstrate that acute stress induces significant changes in autoimmune biomarkers such as CRP, C4, and PEDF. Importantly, recovery-phase biomarker dynamics revealed closer relationships between cortisol and immune markers, suggesting that adaptive responses may emerge during recovery rather than during the stress event itself. Cluster analysis of immune markers demonstrated utility in exploring trends across individual immune profiles.

The identification of biomarker patterns associated with adaptive or maladaptive responses may inform targeted interventions aimed at improving health outcomes in high-stress populations. The insights gained may inform strategies to enhance stress resilience and mitigate autoimmune risk among first responders and other high-stress populations.

## Figures and Tables

**Figure 1 ijms-26-03945-f001:**
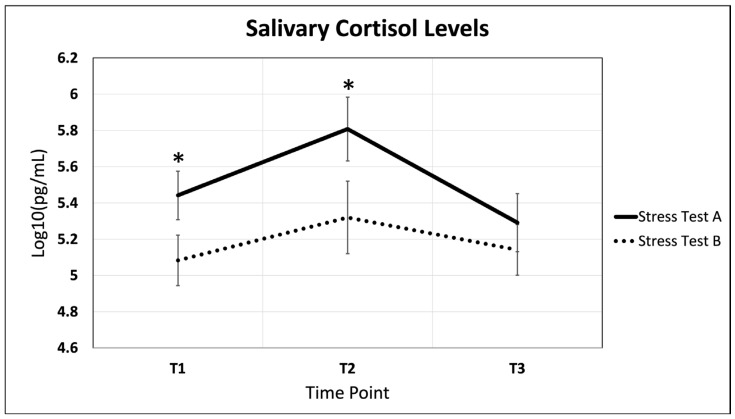
Salivary cortisol trends across both acute stress simulations. Cortisol aggregates increase during acute stress event and decrease during recovery periods. Time points are pre-stress (immediately before stress test), post-stress (immediately following stress test), and recovery (1 h after stress test). Statically significant changes between stress test A and B are displayed with an *. Representative error bars are +/− standard error of the mean. N = 26.

**Figure 2 ijms-26-03945-f002:**
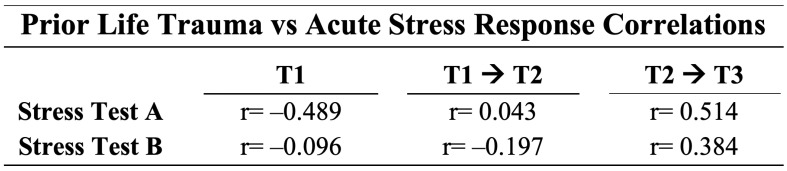
Prior life trauma shows negative correlations at baseline cortisol levels and positive correlations with cortisol movement during recovery time periods. R-values displayed from Pearson correlation coefficients between prior life trauma and cortisol at various time points. Prior life trauma was evaluated using the Life Events Checklist (LEC-5). T1 column was evaluated using log base 10 of cortisol at T1. Columns two and three represent the percentage change in cortisol from T1 to T2 and T2 to T3, respectively. N = 26.

**Figure 3 ijms-26-03945-f003:**
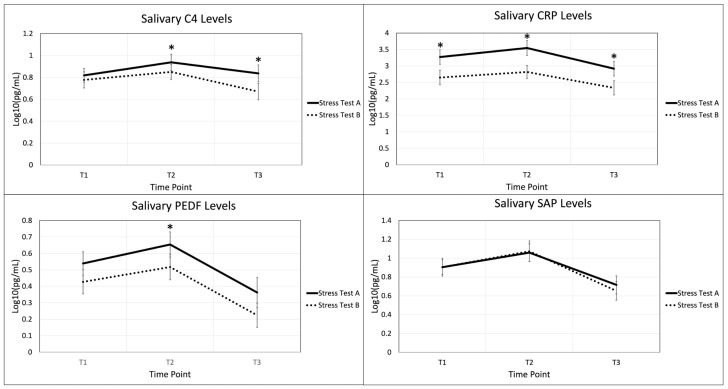
Salivary autoimmune biomarker trends across both acute stress simulations. Aggregates increase during acute stress event and decrease during recovery periods. Time points are (T1) pre-stress (immediately before stress test), (T2) post-stress (immediately following stress test), and (T3) recovery (1 h after stress test). Statically significant changes between stress test A and B are displayed with an *. Representative error bars are +/− standard error of the mean. N = 26.

**Figure 4 ijms-26-03945-f004:**
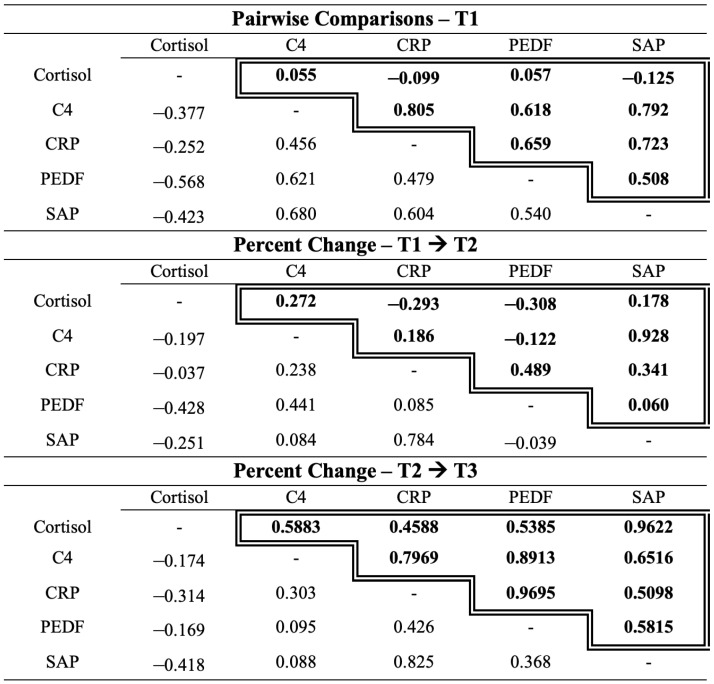
Pair-wise Pearson Correlation Coefficient Matrix for cortisol and the four autoimmune biomarkers throughout the acute stress response for both stress test A and stress test B. Time periods: pre-stress, percentage change during acute stress (T1 → T2), and percentage change during a 1 h recovery (T2 → T3). For each time period, results are separated by stress test: stress test A located in the lower left corner without a border and stress test B located in the upper right corner surrounded by the double-lined border.

**Figure 5 ijms-26-03945-f005:**
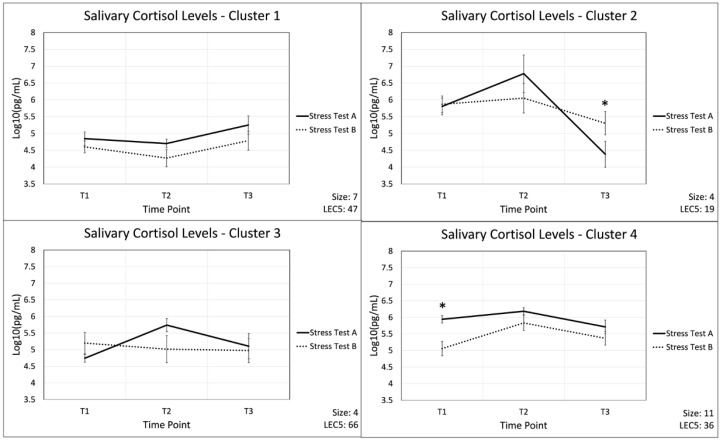
Salivary cortisol trends across both acute stress simulations, separated out into four clusters as determined by K-means algorithm. Aggregates increase during acute stress event and decrease during recovery periods for clusters 2 and 4. Cluster 1 displays a decrease during acute stress event and increase during recovery. The time points are (T1) pre-stress (immediately before stress test), (T2) post-stress (immediately following stress test), and (T3) recovery (1 h after stress test). Statically significant changes between stress test A and B are displayed with an *. Representative error bars are +/− standard error of the mean. N = 26; Cluster 1: 7, Cluster 2: 4, Cluster 3: 4, Cluster 4: 11.

## Data Availability

The original contributions presented in this study are included in the article/Appendix A. Further inquiries can be directed to the corresponding author.

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
