# Peer review of "Acute Stress and Autoimmune Markers: Evaluating the Psychoneuroimmunology Axis in Firefighter Recruits"

_ijms, 2025, doi:10.3390/ijms26093945_

Round 1
Reviewer 1 Report
Comments and Suggestions for Authors
The authors investigated the acute stress effect on autoimmune markers in Firefighter Recruits, evaluating the psychoneuroimmunology axis. The paper also investigated previous trauma interferences in response to stress.The manuscript is well-written, the topic is relevant, and it adds new findings to the literature. However, I have some concerns described in the comments below:
- Please check all the acronyms in the text. Some are in the text without a description of their meaning.
- The introduction is very well written and contains all the information necessary to understand the study, and the hypothesis is straightforward.
- In Figures 1 and 3, place the asterisk symbolizing the statistical significance in the curve, not on the time representation (T1, T2, and T3). Furthermore, remove the methodology from the caption.
- The letters on the graphs in Figure 3 are too light and almost unreadable. Please improve the formatting of the graphs.
- The authors mention in the methodology that they used a questionnaire to assess previous trauma. Please insert the questionnaire used in the supplementary material.
- Regarding the study exclusion criteria. Did the research participants use antidepressants or anxiolytics and hypnotics? Did the participants use anti-inflammatories or immunosuppressors? These drugs during the study may interfere with the results found. Please improve the exclusion criteria for the study in the methodology, making the profile of the participants clear.
- Moreover, only one woman participated in the study. Many studies have shown marked differences between the sexes in the stress response. Why did the authors choose to include a single female representative in the study?
- Add to the methodology the brand and model of the kits used to investigate cortisol levels and the biomarkers CRP, C4, PEDF, and SAP, briefly describing the assay.
- The discussion and conclusions are appropriate, and the authors adequately reported the study's limitations.
Reviewer 2 Report
Comments and Suggestions for Authors
The introduction is highly dense with background information. While the connection between stress and autoimmune markers is well-explained, the introduction could be streamlined to focus more on the study's specific hypotheses and research questions.
Some correlation values are weak or inconsistent across tests, raising questions about the reproducibility of findings.
Some concepts, such as habituation and sensitization, are explained multiple times in different sections, leading to redundancy.
While acknowledging the limitations of cluster analysis, the discussion still draws implications that could be misinterpreted as causal rather than associative.
The discussion frequently justifies limitations by emphasizing the pilot nature of the study, but this does not fully address issues such as sample bias.
While the immune marker findings are intriguing, their significance remains unclear, and more caution could be taken in linking them to autoimmune risk.
The authors largely focus on their primary hypothesis and findings, with limited exploration of opposing interpretations.
The discussion acknowledges the homogeneous sample but does not discuss how gender or racial differences could impact stress and immune responses.
Participants are all from a single fire academy, potentially limiting external validity.
A cohort of 26 participants limits statistical power and generalizability.
No mention of controlling for circadian rhythm effects on cortisol.
Possible variation in hydration levels or recent food intake affecting salivary samples.
Previous sleep patterns or chronic stress levels not considered.
LEC-5 only assesses past trauma at baseline, lacking longitudinal tracking of psychological state changes.
Protease inhibitors were used, but potential degradation before freezing is not discussed.
No comparison with non-firefighter participants or those undergoing different types of stress tests. A control group could help distinguish effects specific to firefighting stress.
Round 2
Reviewer 1 Report
Comments and Suggestions for Authors
The authors reviewed and adequately responded to all questions, making the necessary modifications to improve the paper.
Author Response
Thank you very much for taking the time to review our manuscript, we appreciate your feedback.
Reviewer 2 Report
Comments and Suggestions for Authors
Comment 2
The addition of a limitation is appropriate, but no acknowledgment of specific weak correlations or an attempt to contextualize them is included. Add a short justification, e.g., why some weak correlations are expected in exploratory or pilot studies.
Comment 6
This is a key scientific issue. While the added sentence is helpful, the response could benefit from a more cautious tone in the revised text or note that this area requires further study.
Comment 7
The response adds only one generic sentence, which is insufficient. Include an example or two of alternative interpretations in the discussion section.
Comment 14
The updated text is strong, but the response could highlight how this change improves the study’s transparency or suggests future research needs.
Comment 16
The authors mention this in two places, which is great. The response could be more concise and emphasize how this feedback shapes future directions.
